# Effect of Replacing Fine Aggregate with Fly Ash on the Performance of Mortar

**DOI:** 10.3390/ma16124292

**Published:** 2023-06-09

**Authors:** Dongsheng Zhang, Shuxiang Zhang, Qiuning Yang

**Affiliations:** 1School of Civil and Hydraulic Engineering, Ningxia University, Yinchuan 750021, China; dongsheng.zhang@kuleuven.be (D.Z.); zsxiang4531@163.com (S.Z.); 2Research Group RecyCon, Department of Civil Engineering, KU Leuven, Campus Bruges, 8200 Bruges, Belgium

**Keywords:** fly ash, river sand, mortar, mechanical property

## Abstract

Natural river sand resources are facing depletion, and large-scale mining pollutes the environment and harms humans. To utilize fly ash fully, this study used low-grade fly ash as a substitute for natural river sand in mortar. This has great potential to alleviate the shortage of natural river sand resources, reduce pollution, and improve the utilization of solid waste resources. Six types of green mortars were prepared by replacing different amounts of river sand (0, 20, 40, 60, 80, and 100%) with fly ash and other volumes. Their compressive strength, flexural strength, ultrasonic wave velocity, drying shrinkage, and high-temperature resistance were also investigated. Research has shown that fly ash can be used as a fine aggregate in the preparation of building mortar, thereby ensuring that green-building mortar has sufficient mechanical properties and better durability. The replacement rate for optimal strength and high-temperature performance was determined to be 80%.

## 1. Introduction

The demand for construction sand has increased sharply with rapid developments in engineering construction [1]. China’s sand and stone production accounts for approximately 35% of the world’s sand and stone production, and it could reach 2.5 × 10^10^ t by 2030 [2]. However, natural river sand is a nonrenewable resource formed over billions of years. The world consumes approximately 40 to 50 billion tons of river sand annually in construction projects alone, an increase of 200% over the past 20 years, leading to a global crisis of river sand resource depletion. To solve the lack of river sand resources, domestic and foreign scholars are searching for building sand alternatives to river sand, such as artificial sand and sea sand [3,4,5]. Artificial sand is the most common material used to replace natural river sand. However, it has problems such as uneven grading, high stone powder content, energy consumption during production, and environmental pollution [6]. Sea sand contains soluble salts that can cause the corrosion of steel bars and pose significant safety hazards to structural safety [7,8]. Therefore, developing and utilizing other resources to replace river sand and solve the shortage of construction sand supply is urgent.

The production of fly ash in China is gradually increasing, with an annual production of approximately 600 million tons [9], resulting in a large amount of stockpiled fly ash that occupies a large area of land and causes ecological damage and environmental pollution. The development and utilization of fly ash resources and their application in engineering practices have important economic and environmental benefits [10]. Extensive research has been conducted on the use fly ash as a supplementary cementitious material to replace cement in mortar and concrete [11,12,13]. However, to ensure the superior performance of concrete, the optimal substitution rate of fly ash in concrete often does not exceed 40%, significantly limiting its large-scale consumption of fly ash [14,15,16]. To overcome the limitations of the conventional utilization technology of fly ash, improve the resource utilization rate of fly ash, and alleviate the current shortage of natural river sand resources, scholars have proposed replacing fine aggregates with fly ash to prepare cementitious materials [17,18,19]. Because fly ash acts as a cementitious material and exerts the volcanic ash effect, and partially exerts its “micro aggregate effect”, the cement-based material that replaces fine aggregate with fly ash has better early strength than the cement-based material poured with fly ash instead of cement.

Currently, research on fly ash as a fine aggregate has mainly focused on concrete. Yin [20] established that mixing fly ash and river sand could optimize the grading of fine aggregates and that the compressive strength of the mixture with 30% fly ash was higher than that of the control group. When the fly ash replacement rate was 30%, the compressive strength of the mixture increased by 28.8%. Rafat Siddique [21] studied the effect of F-grade fly ash on the physical and mechanical properties of concrete at a substitution rate of 10–50%, and established that fly ash within a reasonable particle size range can achieve better physical and mechanical properties than the control group at a substitution rate of 50%. In addition, the compressive strength, splitting strength, and flexural strength modulus of elasticity of CFA perform best when the fly ash substitution rate is 50%. Additionally, Mao et al. [22] established that when the fly ash content was below 40%, the strength of fly ash fine aggregate concrete increased with an increase in fly ash content. The active components of fly ash react fully with the cement hydration product Ca(OH)_2_ to generate a C-S-H gel, which can fill the internal pores of concrete. In addition, fly ash is characterised by small particles and a large specific surface area. Therefore, it can fill the large pores between the cement and the aggregate particles, and exhibit an excellent microaggregate-filling effect, thereby making the concrete dense. Ishimaru [23] investigated the basic performance of low-grade fly ash as a fine aggregate component of ordinary concrete. These results indicated that low-grade fly ash is suitable for concrete structures. As a partial substitute for fine aggregates, fly ash can significantly improve the strength performance of ordinary concrete and be effectively used in structural concrete. Zhang et al. [19] established that the attenuation of concrete quality and dynamic elastic modulus was accelerated under the coupling effect of stress damage and high temperature. However, compared with ordinary concrete, fly ash fine aggregate concrete has better freeze–thaw resistance. Bilir [24] studied the influence of fly ash as a fine aggregate on the mechanical properties of mortar. Studies have shown that the strain capacity of mortar was improved, and the pozzolanic effect of fly ash improved the strength of mortar under a 30% replacement rate of fly ash. Although Bilir [24] studied the effect of fly ash as a fine aggregate on the mechanical properties of mortar, the impact of fly ash as a fine aggregate on mortar performance has not yet been studied sufficiently.

Replacing fine aggregate with fly ash can effectively improve the mechanical properties and durability of concrete; however, studies on the effect of replacing fine aggregate with fly ash on the performance of mortar are still sporadic. Therefore, to explore the possibility of replacing natural river sand with fly ash in mortar, various tests were conducted on the effect of replacing different amounts of river sand with Class F fly ash in the same volume on the performance of the mortar, including compressive strength, flexural strength, ultrasonic wave velocity, drying shrinkage, and high-temperature resistance. By determining the appropriate fly ash dosage, we hope to provide a reference for the application of fly ash as a fine aggregate in mortars.

## 2. Test Plan

### 2.1. Raw Materials

The cement used was Ningxia Saima Factory P.O42.5 ordinary Portland cement, and the fine aggregate was natural river sand with a fineness modulus of 2.76, water absorption of 2.67%, and bulk density and apparent density of 1490 kg/m^3^ and 2660 kg/m^3^, respectively. The size distribution of natural river sand is shown in Figure 1. Water requirement of fly ash tests were performed according to GB/T 1596-2017 [25]. A polycarboxylate-based high-efficiency water-reducing agent is suitable for making fly ash fine-aggregate mortar to achieve the target flowability (190–210 mm). The fly ash was provided by the Ningxia Thermal Power Plant. The physical and chemical properties of fly ash are listed in Table 1. The results of chemical analysis of fly ash showed that the total proportion of its main oxides, SiO_2_, Al_2_O_3_, and Fe_2_O_3_, exceeded 70%, and the CaO content was less than 10%. Therefore, according to the classification standard of ASTM C 618 [26], the fly ash used in this study was classified as Class F fly ash. The particle size distribution and X-ray diffraction (XRD) patterns of the fly ash are shown in Figure 1 and Figure 2, respectively.

### 2.2. Mix Ratio

Based on the volume method, cement was used as the binder, natural river sand and fly ash were used as aggregates, and fly ash and other volumes were used to replace natural river sand to prepare fly ash fine-aggregate mortar. The fly ash replacement rates were 0, 20, 40, 60, 80, and 100%, and they were named FA-0, FA-20, FA-40, FA-60, FA-80, and FA-100, respectively. The specific mix proportions are listed in Table 2.

### 2.3. Sample Preparation

The fine fly ash aggregate mortar should be prepared according to the mix proportions listed in Table 2. First, the water reducer was mixed with water, and a glass rod was used for even mixing. The mixed liquid and cement were added to a mixing pot and stirred for 30 s. Subsequently, sand and fly ash were added and mixed for 2 min. The slurry was stirred at a high speed for another 2 min, and then poured into the mould. After the slurry was left to stand for 1 d, the specimens were subjected to standard curing (20 ± 2 °C, relative humidity above 95%) until reaching the measured age.

### 2.4. Test Methods

#### 2.4.1. Compressive Strength and Flexural Strength

Cement mortar strength tests were performed according to GB/T 17671-2021 [27]. The compressive and flexural strength at 1 d, 3 d, 7 d, 28 d, and 90 d were measured. The compressive strength was uniformly loaded at a rate of (2400 ± 200) N/s until failure occurred. The bending strength test was uniformly loaded at a rate of (50 ± 10) N/s until fracture.

#### 2.4.2. Ultrasonic Pulse Velocity

Ultrasonic measurements were performed according to ASTM C 597 [28]. The aged specimens were removed from the standard curing room and wiped with a towel to dry their surfaces. An HC-U81 concrete ultrasonic detector (Beijing Haichuang High Tech Technology Co., Ltd., Beijing, China) was used to measure the propagation speed of ultrasonic pulse waves in the mortar, as shown in Figure 3.

#### 2.4.3. Drying Shrinkage

The drying shrinkage test of the mortar refers to GB/T 29417-2012 [29] using a 40 mm × 40 mm × 160 mm mould. After demoulding the specimen, the initial length of the mortar specimen was measured and then it was placed in an environment with a temperature of 20 ± 2 °C and a relative humidity of 60% ± 5% for drying shrinkage. The length change of the mortar specimens aged for 1, 3, 7, 14, 21, 28, 56, and 90 days was measured using a length comparator. Each group of ratios was tested on three test blocks to obtain an average value.

#### 2.4.4. High Temperature Experiment and Loading Method

After reaching the age limit, the mortar was placed in an oven at 65 °C for 24 h, and then a high-temperature test was conducted using a box resistance furnace. Considering four temperature levels of room temperature, 200, 400, 600, and 800 °C, the heating rate was 5 °C/min. After reaching the target temperature, the temperature remained constant for 2 h; the heating curves are shown in Figure 4. The mass and residual compressive strengths were measured after cooling to room temperature.

## 3. Results and Discussion

### 3.1. Compressive Strength

Compressive strength is an important indicator for evaluating the mechanical properties of materials [30]. The changes in the compressive strength of the fly ash fine aggregate mortar at different curing ages are presented in Figure 5 and Table 3. Figure 5 shows that the compressive strengths of FA-0, FA-20, FA-40, FA-60, FA-80, and FA-100 after 1 d were 13.5 MPa, 13.7 MPa, 13.9 MPa, 14.1 MPa, 14.7 MPa, and 13.9 MPa, respectively. The test results showed that the early strength of the fly ash fine aggregate mortar did not significantly improve. This may be because the early strength of mortar is mainly provided by the hydration products of cement, and the secondary hydration reaction of 1 d fly ash was incomplete. Fly ash plays a significantly greater role in the microaggregate effect than volcanic ash does in the early stages of cement hydration [22]. The strength of the fly ash fine aggregate mortar first increased and then decreased with increasing fly ash substitution rate; however, it was higher than that of the benchmark group. When 80% of the fine aggregate was replaced with fly ash, the compressive strength of the mortar reached its maximum at various ages, with 56.2 MPa and 62.9 MPa at 28 d and 90 d, respectively, which were 23.4% and 23.5% higher than those of the reference group. This was mainly because fly ash has a smaller particle size than natural river sand, which plays a role in filling voids. When the replacement rate of fly ash was 80%, the optimization effect of fly ash on the grading of fine aggregates played a dominant role in improving the strength of the mortar [31,32]. When fly ash replaced fine aggregate by 100%, its mortar strength slightly decreased, reaching 53.2 MPa and 59.4 MPa, decreases of 5.3% and 5.5%, respectively. The lack of natural river sand weakened the bonding strength of the mortar matrix, decreasing the mortar strength [33]. However, this was because of the excessive addition of fly ash after 100% substitution, which weakened the cement hydration reaction rate [34]. Therefore, owing to the microaggregate effect, the compressive strength of fly ash mortar was superior to that of ordinary mortar, with an optimal dosage of 80%.

### 3.2. Flexural Strength

The variation pattern of the flexural strength of the fly ash fine aggregate mortar is presented in Figure 6 and Table 4. Compared with the benchmark group, the flexural strength of the fly ash fine aggregate mortar increased by 1.1, 2.8, 3.5, 4.9, and 1.4% at the 7-day age at 20, 40, 60, 80, and 100% substitution rates, respectively; at 28 d, the flexural strength of the mortar increased by 5.8, 9.4, 13.9, 16.1, and 7.4%, respectively. It can be observed that under different dosages, the increase in the 28 d flexural strength was greater than that at 7 d, which could be related to the pozzolanic effect of fly ash. Fly ash reacts twice with the cement hydration product, Ca(OH)_2_, to generate a new C-S-H gel, which improves the strength [35]. In addition, the flexural strength of the mortar first increased and then decreased with the increasing fly ash substitution rate; however, all rates were higher than those of the reference group, which was consistent with the compressive strength law. Similarly, when the replacement rate of fly ash was 80%, the flexural strength of the mortar reached its maximum at all ages, with values of 3.6, 9.2, 9.9, 12.0, and 12.5 MPa at 1, 3, 7, 28, and 90 d, respectively, which were 2.9, 11.7, 4.9, 16.1, and 14.7% higher than those of the reference group, respectively. Although the mortar strength decreased slightly compared to FA-80 when 100% of the fine aggregate was replaced, the flexural strength of the 3, 7, 28, and 90 d groups increased by 1.0, 6.1, 1.4, 7.4, and 7.4, respectively, compared to those of the reference group.

### 3.3. Relationship between the Compressive Strength and Flexural Strength of Mortar

There is a close relationship between compressive strength and flexural strength. The formula for calculating the flexural strength of ordinary concrete given by ACI is:(1)$$f_{\mathrm{tf}}=0.81\sqrt{f_{\mathrm{cu}}},$$
where f_tf_ is the flexural strength of concrete; f_cu_ is the compressive strength of concrete.

The recommended conversion formula for f_tf_ and f_cu_ in the European Norms (CEB) is:(2)$$f_{\mathrm{tf}}=0.54\sqrt{f_{\mathrm{cu}}}$$

The recommended conversion formula for f_tf_ and f_cu_ in the Indian standard (IS 456) [36] is:(3)$$f_{\mathrm{tf}}=0.7\sqrt{f_{\mathrm{cu}}}$$

However, the formula for calculating the flexural strength of ordinary concrete does not accurately reflect the effect of fly ash on fine aggregate mortar. Therefore, it is necessary to modify the conversion formula for the flexural strength. Based on the experimental results, Figure 7 shows the correlation between the flexural and compressive strength of the fly ash fine aggregate mortar. Through regression analysis, it is recommended that the formula for calculating the flexural strength of fly ash fine aggregate mortar is:(4)$$f_{\mathrm{tf}}=0.573f_{\mathrm{cu}}^{0.752}$$

The correlation coefficient R^2^ obtained by regression was 0.972, indicating that the experimental dispersion was relatively small, owing to the uniformity of the fly ash fine aggregate mortar.

### 3.4. Ultrasonic Pulse Velocity

Ultrasonic pulse testing (UPV) can be used to evaluate the structural development, internal cracks, and microstructures of cement-based materials [37]. The UPV test results for fly ash fine aggregate mortar with different substitution rates are presented in Figure 8. The ultrasonic pulse initially increased and then decreased with increasing fly ash content. At the 80% substitution rate, the ultrasonic pulse wave velocities of the 1 d, 3 d, 28 d and 90 d mortar reached their maximum at 3.1 km/s, 3.6 km/s, 4.0 km/s and 4.3 km/s, respectively. This is because the reaction between the fly ash and the hydration product Ca(OH)_2_ of cement consumes water, and the hydration product fills the internal pores of the mortar, making the specimen denser [38]. The velocity of ultrasonic pulses is influenced by the medium, and their propagation speeds in solids are higher than those in liquids and air. Therefore, the pulse velocity first increased and then decreased with increasing fly ash content [39]. In addition, the ultrasonic pulse velocity increased with an increase in curing age because the internal density of the specimen is higher in mortars with higher strengths; thus, the pulse velocity was higher. The pulse velocity of the mortar ranged from 2.9 to 3.1 km/s for 1 d and 4.1 to 4.3 km/s for 90 d, indicating that the age of the mortar had a significant effect on the ultrasonic pulse velocity. This is because in the early mortar stage, owing to an insufficient hydration reaction, more free water is present, and the internal pores are relatively large [40]. As the age increases, the internal cement hydration reaction became more thorough. The cement hydration and secondary hydration reactions of fly ash consume water and the hydration products fill the internal pores of the mortar, rendering the specimen denser [41].

As shown in Figure 9, there was a certain correlation between the propagation speed of ultrasonic waves in fly ash fine aggregates and their compressive strength. Through regression analysis, it can be concluded that:(5)$$f_{\mathrm{cu}}=33.81x_{v}-86.63$$
where x*_v_* is the ultrasonic pulse velocity, and f_cu_ is the compressive strength of the concrete. The correlation coefficient R^2^ obtained using the regression formula was 0.958. Establishing a regression equation can provide a technical reference for detecting the strength of fly ash fine aggregate mortars, using the ultrasonic method.

### 3.5. Drying Shrinkage

Mortar and concrete are prone to dry shrinkage deformation in low-humidity environments, accelerating shrinkage cracking and resulting in poor durability and reduced service life and safety. The particle shape, strength, and interface structure between the aggregate and matrix are the key factors constraining mortar shrinkage. In addition to having certain mechanical properties, fly ash fine aggregate mortar should have good volume stability. The changes in the lengths of the fly ash fine aggregate mortar caused by drying shrinkage are shown in Figure 10. The drying shrinkage of the mortar increased rapidly within 42 d, with the most significant increase in the first 7 d, and a decrease in the shrinkage rate after 42 d. This was because after 42 d the drying shrinkage rate improved owing to the increase in hydration products and decrease in water evaporation [41]. As the replacement rate of fly ash increased, the anti-shrinkage ability of the mortar gradually increased. At the 100% substitution level, the mortar had the optimal drying shrinkage property and the drying shrinkage rate was reduced by 29.0% compared with FA-0. This was mainly owing to the activity and micro-aggregation effects of fly ash. The activity effect of fly ash was that it reacted with the Ca(OH)_2_ produced by cement hydration to produce hydrated calcium silicate and hydrated calcium aluminate gel, which filled some internal voids, reduced the internal porosity of the mortar, and limited its shrinkage [42]. In addition, fly ash could effectively replace fine aggregates, resulting in a micro-agglomeration effect. The fly ash contained many small glass beads and debris particles that were evenly distributed inside the mortar. In the mixed slurry, some of the pores were filled with these particles, which significantly enhanced the internal compactness of concrete and limited its shrinkage considerably [43].

### 3.6. Formatting of Mathematical Components

After the mortar underwent high-temperature treatment, various changes occurred, resulting in varying degrees of mass loss. Moreover, the quality loss and its causes varied depending on the temperature. Figure 11 shows the mass loss rate of the fly ash fine aggregate mortar at different temperatures. As shown in Figure 11, as the temperature increased, the mass loss rate of the mortar gradually increased; however, the effect of the fly ash replacement rate on the mass loss was not significant. The mass loss rates of mortar at 200, 400, 600, and 800 °C were approximately 7.4, 9.7, 11.9, and 16.2%, respectively. At temperatures of 400 °C and below, significant losses in mortar mass occurred owing to the evaporation of free water and cementitious water inside the mortar, as well as the loss of bound water caused by the dehydration of hydration products [41]. Previous studies have demonstrated that [44] Ca(OH)_2_ decomposes in large quantities at 500 °C, whereas C-S-H begins to decompose at around 600 °C, resulting in a continued increase in mortar quality loss [45]. At 800 °C, the mass loss rate of mortar increased significantly, again, mainly owing to the extensive decomposition of C-S-H [46].

Figure 12 and Table 5 show the changes in the compressive strength of the mortar at different high temperatures. As shown in Figure 12, the compressive strength at each substitution level first slightly increased and then decreased as the action temperature increased, and the compressive strength of the fly ash fine aggregate mortar was higher than that of the reference group. When the operating temperature was 200 °C, the strength of the mortar slightly increased. At 200 °C, compared with FA-0, the compressive strength of FA-20, FA-40, FA-60, FA-80, and FA-100 mixtures increased by 4.0, 10.1, 18.2, 22.5, and 16.7%, respectively. An increase in temperature from room temperature to 200 °C considerably promoted sufficient hydration inside the concrete, filling internal pores with hydration products and making the internal structure more dense, resulting in an increase in the compressive strength of the specimen within this temperature range [47]. At 400 °C, some hydration products such as Ca(OH)_2_ and C-S-H in the mortar begin to decompose, and the compressive strength decreased slightly [48]. At 600 °C and 800 °C, the compressive strength significantly decreased and the internal cementitious material continued to dehydrate and decompose, leading to the loss of interlayer water and chemically bound water and causing damage to the internal structure of the mortar [49]. At 800 °C, compared with FA-0, the compressive strength of FA-20, FA-40, FA-60, FA-80, and FA-100 mixtures increased by 9.7, 55.3, 111.4, 114.0, and 140.4%. The decrease in the compressive strength of the sample replaced by fly ash at 800 °C was significantly smaller than that of the ordinary mortar. The most important reason was that the microaggregate effect of fly ash played an effective role in making the mortar high-temperature resistant [50]. However, fly ash can undergo a secondary reaction with the cement hydration product Ca(OH)_2_ to generate new hydrated calcium silicate. The hydration product can fill the pores, forming a dense structure and improving the mortar strength [51].

## 4. Conclusions

In this study, river sand with different fly ash ratios (0, 20, 40, 60, 80, and 100%) was used to study the compressive strength, flexural strength, ultrasonic wave velocity, drying shrinkage and high temperature of mortar. Based on the research results, the following conclusions were drawn:With increasing fly ash content, the strength of the fly ash fine aggregate mortar first increased and then decreased. When the fly ash content was 80%, the mortar strength reached its maximum, with a compressive strength of 62.93 MPa and a flexural strength of 12.47 MPa at 90 d, which were 23.56% and 14.72% higher than those of the ordinary mortar, respectively. This indicates that fly ash can be used as a fine aggregate to prepare green building mortar with mechanical properties that meet application requirements and have good engineering application values.The correlation between the compressive and flexural strengths of the fly ash fine aggregate mortar was corrected by comparing the specifications for different regions. The revised correlation model between the compressive and flexural strengths was more accurate. In addition, the variation trends of the ultrasonic pulse wave velocity and flexural compressive strength were the same, with a maximum of 4.23 km/s reached when the fly ash content was 80%.The results of the drying shrinkage test indicated that the higher the amount of fly ash replacing the fine aggregate, the smaller the drying shrinkage strain of the mortar. The addition of fly ash significantly affected the 90 d drying shrinkage rate. At the 100% substitution level, the drying shrinkage property of the mortar reached its optimal level, and the drying shrinkage rate was reduced by 29.01% compared to that of ordinary mortar. The drying shrinkage results also indicated that fly ash inhibited the drying shrinkage of the mortar.The high-temperature test results indicated that using fly ash as a substitute for natural river sand can improve the high-temperature resistance of mortars. Mortars containing 80% fly ash as a substitute for natural river sand had a higher residual compressive strength than the other mortars. In addition, the addition of fly ash did not significantly affect the quality loss of the mortar at high temperatures.

## Figures and Tables

**Figure 1 materials-16-04292-f001:**
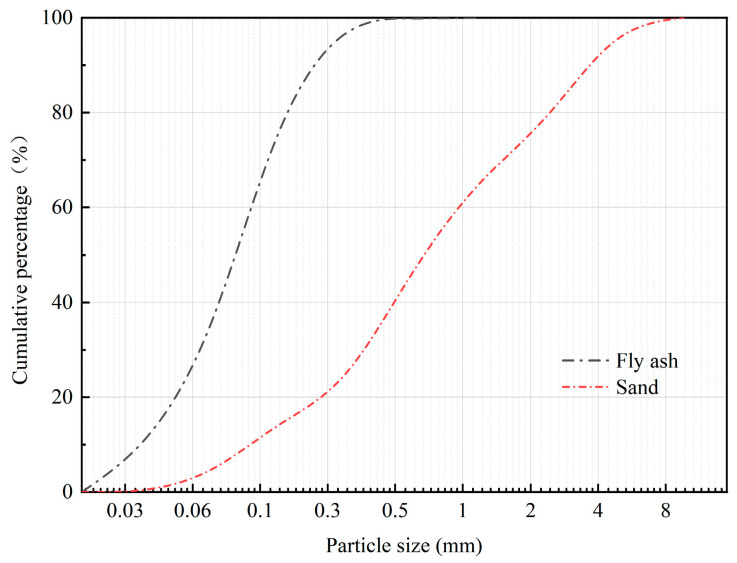
Particle size distribution of fly ash and sand.

**Figure 2 materials-16-04292-f002:**
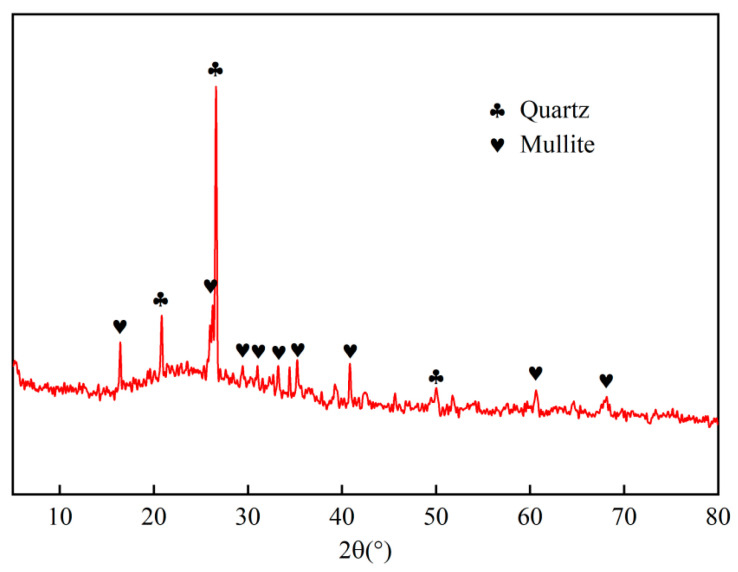
XRD patterns of fly ash.

**Figure 3 materials-16-04292-f003:**
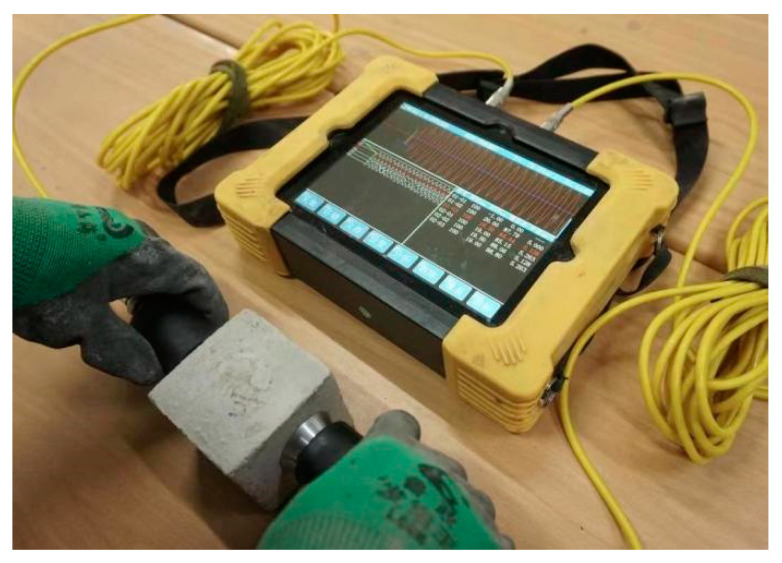
Ultrasonic wave velocity test chart.

**Figure 4 materials-16-04292-f004:**
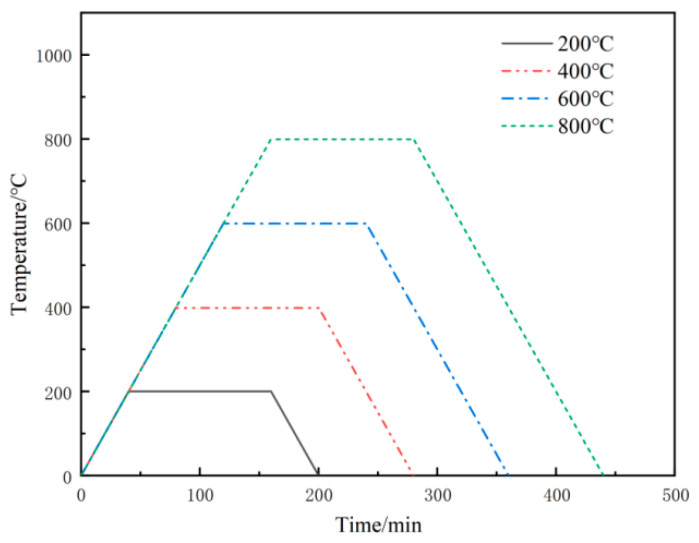
Heating curves.

**Figure 5 materials-16-04292-f005:**
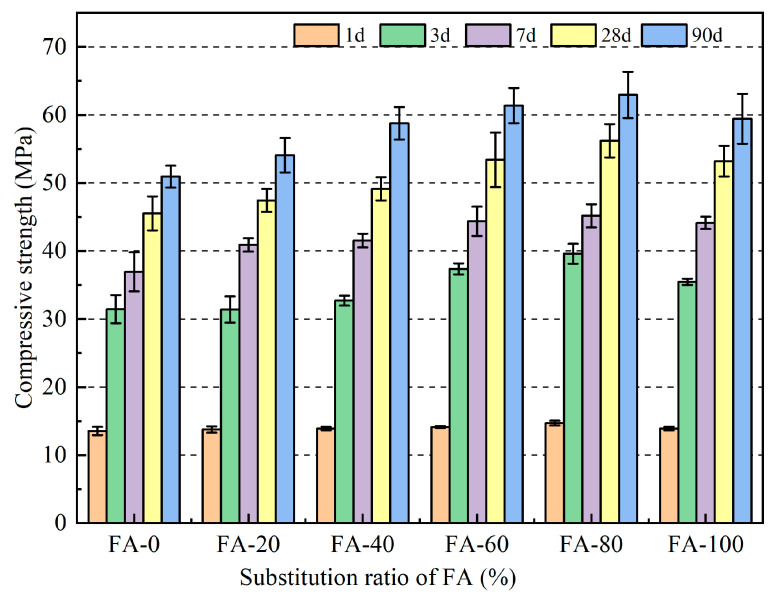
Compressive strength of fly ash fine aggregate mortar.

**Figure 6 materials-16-04292-f006:**
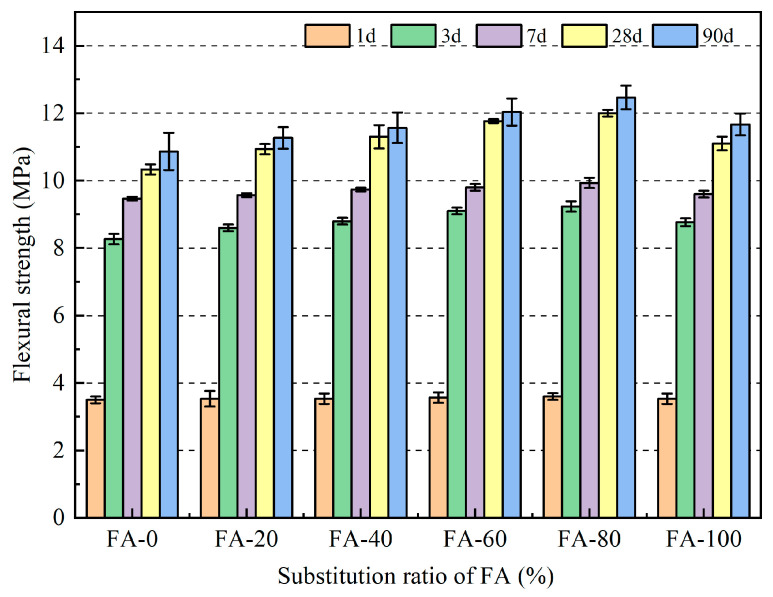
Flexural strength of fly ash fine aggregate mortar.

**Figure 7 materials-16-04292-f007:**
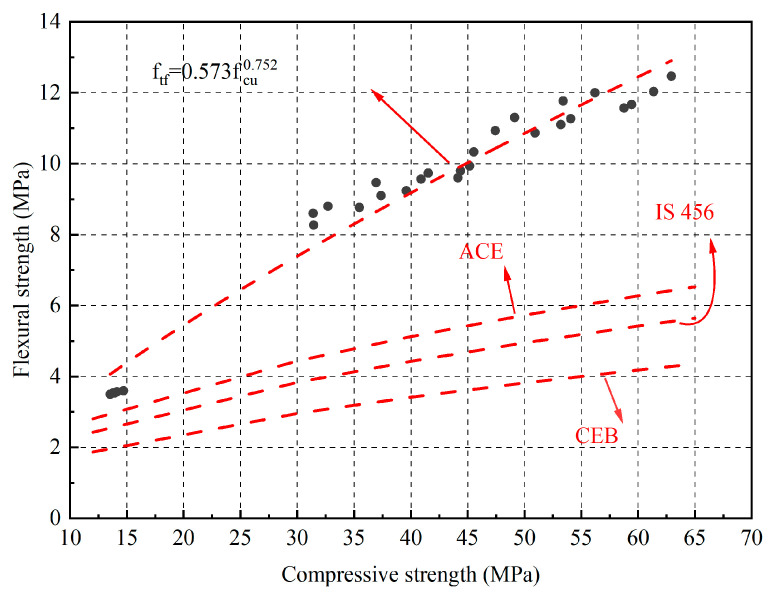
The relationship between the flexural strength and compressive strength of mortar.

**Figure 8 materials-16-04292-f008:**
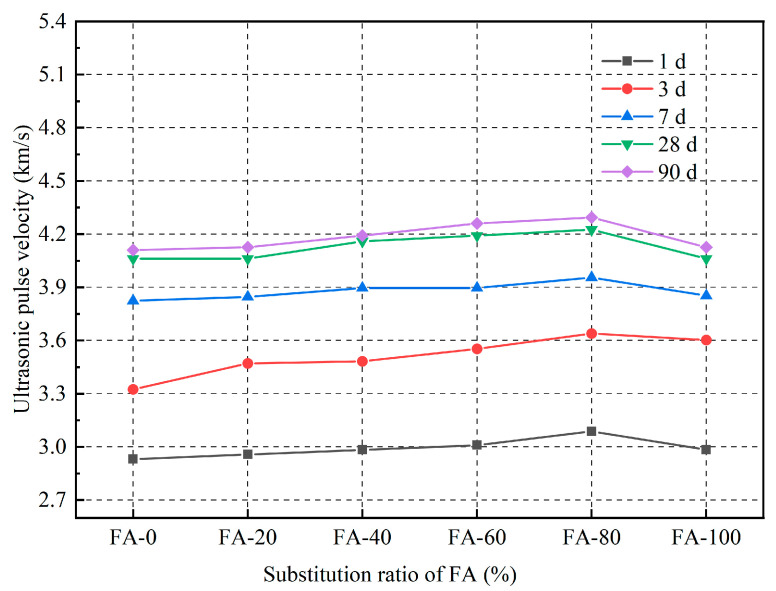
Ultrasonic pulse velocity of fly ash fine aggregate mortar.

**Figure 9 materials-16-04292-f009:**
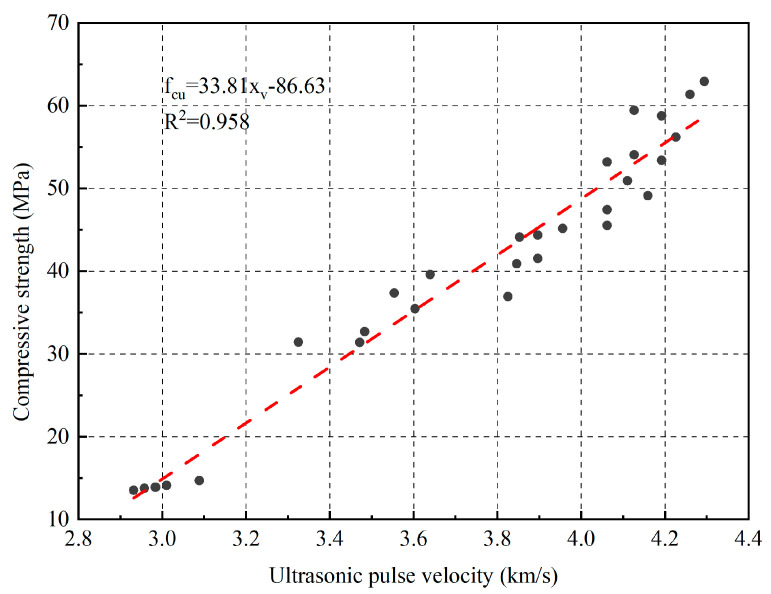
The relationship between compressive strength and ultrasonic pulse velocity of mortar.

**Figure 10 materials-16-04292-f010:**
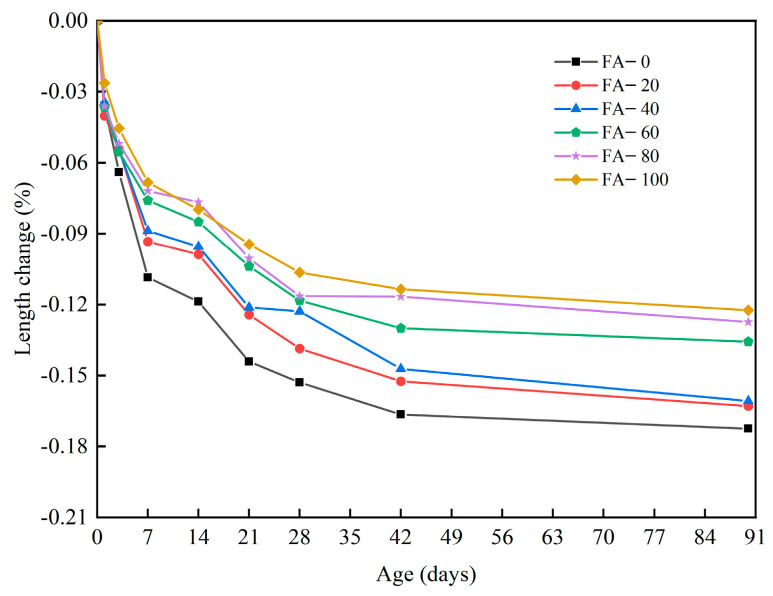
Drying shrinkage of the fly ash fine aggregate mortar.

**Figure 11 materials-16-04292-f011:**
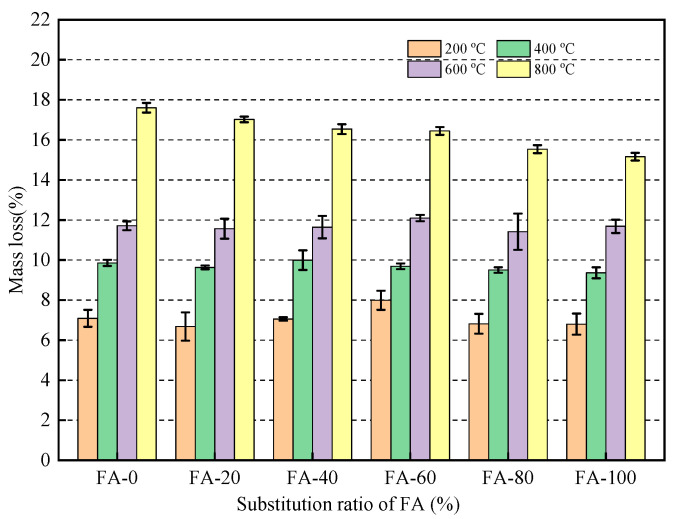
Mass loss of mortar after high temperature treatment.

**Figure 12 materials-16-04292-f012:**
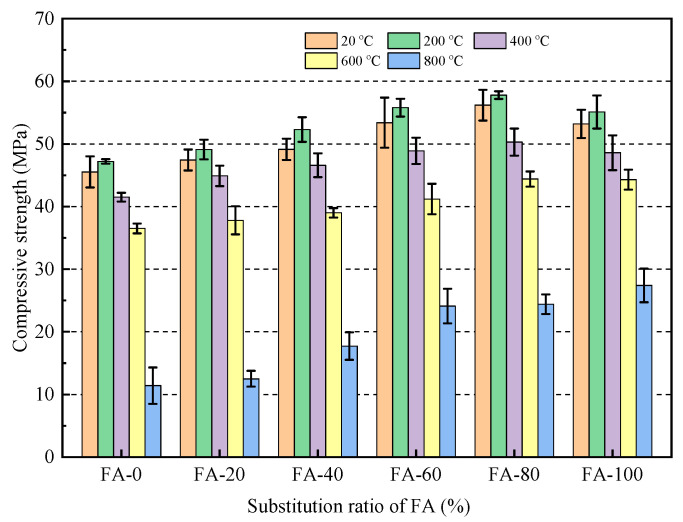
Compressive strength of mortar after treatment at different high temperatures.

**Table 1 materials-16-04292-t001:** Properties of fly ash.

Chemical Composition	wt%	Physical Properties	Value
SiO_2_	50.35	Density (g·cm^−3^)	2.13
Al_2_O_3_	29.65	Water requirement (%)	97
Fe_2_O_3_	6.61	Loss on ignition (%)	6.3
CaO	5.85	45 μm sieve remaining (wt.%)	37.1
ZrO_2_	0.11		
SrO	0.17		
MnO	0.10		
P_2_O_5_	1.13		
TiO_2_	1.75		

**Table 2 materials-16-04292-t002:** Mixture proportions of the mortars.

Content (%)	Water/kg	Cement/kg	Sand/kg	Fly Ash/kg	Superplasticiser/kg	Fluidity/mm
FA-0	294	700	1120	0	3.3	203
FA-20	294	700	896	172.5	3.7	199
FA-40	294	700	672	345.1	4.3	208
FA-60	294	700	448	517.6	4.7	197
FA-80	294	700	224	690.2	5.1	209
FA-100	294	700	0	862.7	6.4	206

**Table 3 materials-16-04292-t003:** Relative change in the compressive strength of mortar (%).

Code	1 d	3 d	7 d	28 d	90 d
FA-0	101.7	100.0	100.0	100.0	100.0
FA-20	102.7	99.9	110.7	104.2	106.6
FA-40	104.4	104.0	112.5	107.9	115.4
FA-60	108.6	118.9	120.1	117.3	120.5
FA-80	102.7	126.0	122.3	123.4	123.6
FA-100	101.7	112.8	119.5	116.8	116.7

**Table 4 materials-16-04292-t004:** Relative change in the flexural strength of mortar (%).

Code	1 d	3 d	7 d	28 d	90 d
FA-0	101.0	100.0	100.0	100.0	100.0
FA-20	101.0	104.0	101.1	105.8	103.7
FA-40	101.9	106.5	102.8	109.4	106.4
FA-60	102.9	110.1	103.5	113.9	110.7
FA-80	101.0	111.7	104.9	116.1	114.7
FA-100	101.0	106.1	101.4	107.4	107.4

**Table 5 materials-16-04292-t005:** Changes in the compressive strength of mortar after different high temperatures (%).

Code	20 °C	200 °C	400 °C	600 °C	800 °C
FA-0	100.0	100.0	100.0	100.0	100.0
FA-20	104.2	104.0	108.2	103.6	109.7
FA-40	107.9	110.8	112.3	106.7	155.3
FA-60	117.3	118.2	117.8	112.9	211.4
FA-80	122.7	122.5	121.2	121.6	214.0
FA-100	118.3	116.7	117.1	121.4	240.4

## Data Availability

Not applicable.

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
