# Peer review of "Effect of Replacing Fine Aggregate with Fly Ash on the Performance of Mortar"

_materials, 2023, doi:10.3390/ma16124292_

Round 1
Reviewer 1 Report
This study used low-grade fly ash as a substitute for natural river sand in mortar. Some suggestions to consider to improve the quality of the article are as follows:
- In the introduction, it should be more discussion on the studies on the mechanism of using fly ash as a fine aggregate to replace natural sand, and how is the aggregate alkaline reactivity?
- It should clarify the concept of how fly ash is called low quality.
- How is the coal burning technology of the fly ash used in this study (Circulating Fluidized Bed Fly Ash- CFB or pulverized coal combustion- PCC)? Some material properties should be added as the particle composition, Water absorption of natural sand and fly ash; the activity of the used fly ash? Need to supplement the experiment to evaluate the alkali reactivity to confirm the ability to use fly ash as a substitute for natural sand in mortar production?
- Table 3, and Table 4 Should be shown as the line plot with a time scale to clearly the strength development over time of the different fly ash contents.
- Regarding the Ultrasonic pulse velocity of the fly ash fine aggregate mortar results, why only test on samples at 3 and 28 curing days? What are the sample test conditions (wet/dry)? Consider drawing a graph of the relationship between the Ultrasonic pulse velocity and the compressive strength of the mortar sample? It should be explained why when there is fly ash 20-100%, the Ultrasonic pulse velocity does not change significantly.
some technical words should be edited: "machine-made" to "artificial sand"; Water demanded; The mixed mortar was "injected" into the mold to "poured"; ...
Reviewer 2 Report
The manuscript reports on the basic characteristics of common cement-based mortars with partial or even full replacement of sand aggregate by fly ash. It is highly questionable whether the replacement of relatively coarse aggregate (river sand) by very fine fly ash is a reasonable approach. The end member is in fact a cement paste with quite high amount of fly ash and not mortar. Based on these aspects, the scientific soundness of the manuscript is low, without deep investigations and understanding of the observed phenomena. The manuscript is quite readable in general but needs careful revisions at specific places. If editor finds this manuscript suitable for Materials journal, I recommend considering its publication after its improvements are done.
More detailed information on characteristics of the used materials is necessary. In particular, particle size distribution of all solid components should be specified.
What is the reason for considering the used fly ash as “low-grade”? High amount of unburnt carbon? Please, provide also mineralogical composition of the used fly ash, including the content of amorphous phase.
I would appreciate application of instrumental techniques like XRD, TG-DTA and SEM to strengthen the presented data and their discussion.
Line 70: missing number of the reference
Table 1: What is water demanded (%)? Subscripts in the chemical formula should be used.
Line 110–112: Please, revise.
2.4.1 Compressive strengths after 24 hours should be also included. Early strengths are important from the practical point of view and three days is quite late. It can be expected that there could be different effect on very early strengths due to the increasing plasticizer dose as the amount of fly ash increases.
2.4.2 Please, provide more information on the UPV measurement like frequency and number of readings for each specimen at each age.
150–153: The explanation does not make sense. Moreover, the strengths of FA20 is the same as the reference FA0 after 3 days.
Table 3 and 4 and expression of numbers in general: Writing the values with an accuracy of two decimal places is not meaningful. Many values are the same with respect to the range of error bars. This should be revised over the entire manuscript.
213: Revise the number of the formula. Moreover, its general applicability is speculative, it makes sense only for the presented data.
Figure 5: It could be interesting to show also the prediction using standardized formulas 1, 2, 3 for comparison.
Figure 7: Mass changes during drying should be included.
Please, revise the term “anti-shrinkage ability”.
Conclusions, point 2: Revise the whole first sentence.
Some nonsense sentences, examples included in the comments above.
Round 2
Reviewer 2 Report
I appreciate that most of my comments were reflected during the revisions and recommend the paper for publication in Materials.
Still, it is a bit of a pity that the mass loss during drying were not determined and commented on, because it could be interesting with respect to the proposed changes in porosity and the rate of shrinkage.
Author Response
Thank you for your kind suggestions. We will take note of this in future research.